# Single-cell RNA sequencing highlights the role of inflammatory cancer-associated fibroblasts in bladder urothelial carcinoma

Zhaohui Chen[1,4], Lijie Zhou[1,2,4], Lilong Liu[1,2], Yaxin Hou[1,2], Ming Xiong[1], Yu Yang[3], Junyi Hu[1,2✉] & Ke Chen [1,2✉]

Although substantial progress has been made in cancer biology and treatment, clinical outcomes of bladder carcinoma (BC) patients are still not satisfactory. The tumor micro-environment (TME) is a potential target. Here, by single-cell RNA sequencing on 8 BC tumor samples and 3 para tumor samples, we identify 19 different cell types in the BC micro-environment, indicating high intra-tumoral heterogeneity. We find that tumor cells down regulated MHC-II molecules, suggesting that the downregulated immunogenicity of cancer cells may contribute to the formation of an immunosuppressive microenvironment. We also find that monocytes undergo M2 polarization in the tumor region and differentiate. Furthermore, the LAMP3 + DC subgroup may be able to recruit regulatory T cells, potentially taking part in the formation of an immunosuppressive TME. Through correlation analysis using public datasets containing over 3000 BC samples, we identify a role for inflammatory cancer-associated fibroblasts (iCAFs) in tumor progression, which is significantly related to poor prognosis. Additionally, we characterize a regulatory network depending on iCAFs. These results could help elucidate the protumor mechanisms of iCAFs. Our results provide deep insight into cancer immunology and provide an essential resource for drug discovery in the future.

[1] Department of Urology, Union Hospital, Tongji Medical College, Huazhong University of Science and Technology, Wuhan, Hubei, China. [2] Shenzhen Huazhong University of Science and Technology Research Institute, Shenzhen, China. [3] Department of Pathology and Laboratory Medicine, Indiana University School of Medicine, Indianapolis, IN, USA. [4]These authors contributed equally: Zhaohui Chen, Lijie Zhou. ✉email: M201875542@hust.edu.cn; shenke@hust.edu.cn

Bladder carcinoma (BC) is one of the most prevalent uro-genital malignant diseases around the world, with approximately 430,000 new cases diagnosed and over 165,000 deaths caused per year[1]. Although substantial progress has been made in cancer biology and treatment, clinical outcomes of bladder carcinoma patients are still not satisfactory[2].

In the last decade, the tumor microenvironment (TME) has been a popular area of cancer biology research in relation to therapeutic targets for drug discovery. Notably, BC is one of the least immune infiltrated cancers[3], which may account for the poor response to anti-PD1 therapy. Since then, designing new treatment strategies for BC has continued to be an arduous task.

Previously, molecular subtypes of BC have been reported to show different cell type-specific expression patterns[4], which indicates that the heterogeneity of BC at least partly results from different cell type fractions inside the microenvironment. However, relevant studies have been rare until recently.

In the current study, we profile the transcriptome of 52721 single cells from bladder urothelial carcinoma or para-tumor mucosa samples and produce an atlas of the whole TME inside bladder cancer tissues. Our work highlights the role of one cancer-associated fibroblast (CAF) subset, named inflammatory cancer-associated fibroblasts (iCAFs), in BC and discovers possible therapeutic targets for BC treatment. In addition, we investigate the relation between TME and molecular subtypes of BC by correlating scRNA-seq data to over 3500 bulk RNA sequencing or microarray profiles in public datasets. These results promote the understanding of heterogeneity between patients and provide a basis for individualized treatment for bladder urothelial carcinoma.

## Results

**Single-cell sequencing and cell type identification.** After quality control and removal of the batch effect between batches (Supplementary Fig. 1, see "Methods"), 52721 single cells were clustered into eight major clusters. Cluster-specific genes were used to annotate cell types with classic markers described in previous studies[5]: epithelial (EPCAM+) cells; endothelial (CD31+) cells; two types of fibroblasts (COL1A1+)—iCAFs (PDGFRA+) and myo-CAFs (mCAFs) (RGS5+); B cells (CD79A+); myeloid cells (LYZ+); T cells (CD3D+); and mast cells (TPSAB1+) (Fig. 1a, b, Supplementary Fig. 2A–D).

**Cancer cells show high heterogeneity due to CNV patterns.** EPCAM+ epithelial cells (EPCs) were reclustered to produce 17 clusters (Fig. 1c, d). Interestingly, although the batch effect has been previously removed, cancer cells still showed a patient-specific expression pattern, which indicated extremely high heterogeneity that was probably caused by copy number variations (CNVs)[6]; this assumption was confirmed by InferCNV (Fig. 1c, Supplementary Fig. 3A). As shown in Supplementary Fig. 3A, CNVs accumulated in most high-grade tumor-derived EPCs and showed high heterogeneity among clusters. Despite the heterogeneity, almost all high-grade cancer cells possessed deletions from chromosomes 9 and 11 and amplifications in chromosomes 19 and 20. In addition, some cells from tumor tissue possessed almost no CNV and showed a similar expression pattern to normal EPCs (Fig. 1d), which indicated that these cells may be non-malignant EPCs. Therefore, EPCs were divided into 4 groups based on CNVs: low, high, control, and undetected in this investigation.

When comparing the transcriptomes, we noticed that a series of genes was especially expressed in the control and CNV undetected groups but almost absent in tumor cells (Fig. 1d, red blank). Gene Ontology enrichment analysis revealed that these genes were enriched in immune-related pathways, especially B cell-related pathways (Fig. 1e). Comparing to normal epithelial cells, cancer cells almost lost the ability to produce immunoglobulin, and they expressed lower levels of MHC-II molecules, which was validated by immunofluorescence (Fig. 1f, g). These results indicated that bladder cancer cells might downregulate immunogenicity to escape immune detection. In addition, we noticed that cancer cells possessing more CNVs seemed to express higher levels of IGF2 (Supplementary Fig. 3B, C). In the TCGA BLCA cohort, a high level of IGF2 was significantly related to poor prognosis (Supplementary Fig. 3C). Pathway analysis with gene set variation analysis (GSVA) revealed that E2F targets, MYC targets and the G2M checkpoint pathway were enriched in the CNV high group, while inflammatory and other immune-related pathways were downregulated (Fig. 1h). These results further confirmed that cancer cells in the advanced stage of BC downregulated immunogenicity and displayed high proliferation ability.

**Monocytes recruited into the tumor region experience M2 polarization.** Reclustering of myeloid cells identified seven cell types: tumor-associated macrophages (TAMs, MRC1+C1[high]); CD1C+ dendritic cells (CD1C+ DCs, CD1C+CLEC10A+); monocytes (S100A8+S100A9+); proliferating myeloid cells (TOP2A+); cross-presenting DCs (CLEC9A+); follicular B cells (CD79A+MS4A1+); and LAMP3+ DCs (LAMP3+CCR7+). These cell subgroups were confirmed by flow cytometry (Supplementary Fig. 4A). Two cell clusters expressed both myeloid markers, and epithelial or endothelial markers, and they were considered doublets (Fig. 2a, Supplementary Fig. 5A–C). Differences between cell types could also be identified by the activity of TF motifs (Supplementary Fig. 5D). Notably, monocytes mostly originated from normal mucosal tissues, while TAMs were enriched in BC tissues. In addition, it seems that the transcriptomes of these two cell types exhibited continual changes (Fig. 2b, Supplementary Fig. 5B, red blank), indicating that monocytes recruited into the tumor region were reprogrammed into TAMs, which has been previously reported in a murine breast cancer model[7]. To further investigate this ongoing process, we performed trajectory analysis and RNA velocity analysis on monocytes and TAMs[8,9]. Similar phenomenon was observed in both two computational pipelines (Fig. 2c and Supplementary Fig. 6B). Combined with the key motifs identified by SCENIC, we recognized that the activity of three motifs, BACH1, MAFG, and NFE2, was downregulated, while activation of the MAF, STAT1 and STAT2 motifs led to this M2-polarization process (Fig. 2c, d, Supplementary Fig. 6A). These results provide potential targets for inhibiting or reversing the formation of the immunosuppressive microenvironment. In addition, we noticed that the co-inhibitors CD274, LGALS9, CD276, TIGIT, and PDCD1LG2 were all upregulated in this differentiation process, while co-activators were downregulated (Fig. 2e).

**LAMP3+ DCs recruit Tregs into the tumor region via cytokines.** Among the 3 DC subgroups, LAMP3+ DCs expressed various genes encoding cytokines, including CCL17, CCL19, and CCL22 (Fig. 2f). Strikingly, these cytokines originated almost entirely from LAMP3+ DCs in BC (Fig. 2g). As discussed in a previous study, CCL17 and CCL22 have strong chemotaxis towards Tregs via binding to CCR4 expressed on the membrane of Tregs[10,11]. In the TCGA BLCA cohort, the LAMP3+ DC signature was highly positively correlated with the Treg signature and Th2 signature, which were both CCR4+, but there was not a high correlation with the CTL signature (Fig. 2h). LAMP3+ DCs were significantly enriched in tumor tissues (Supplementary Fig. 5A). Additionally,

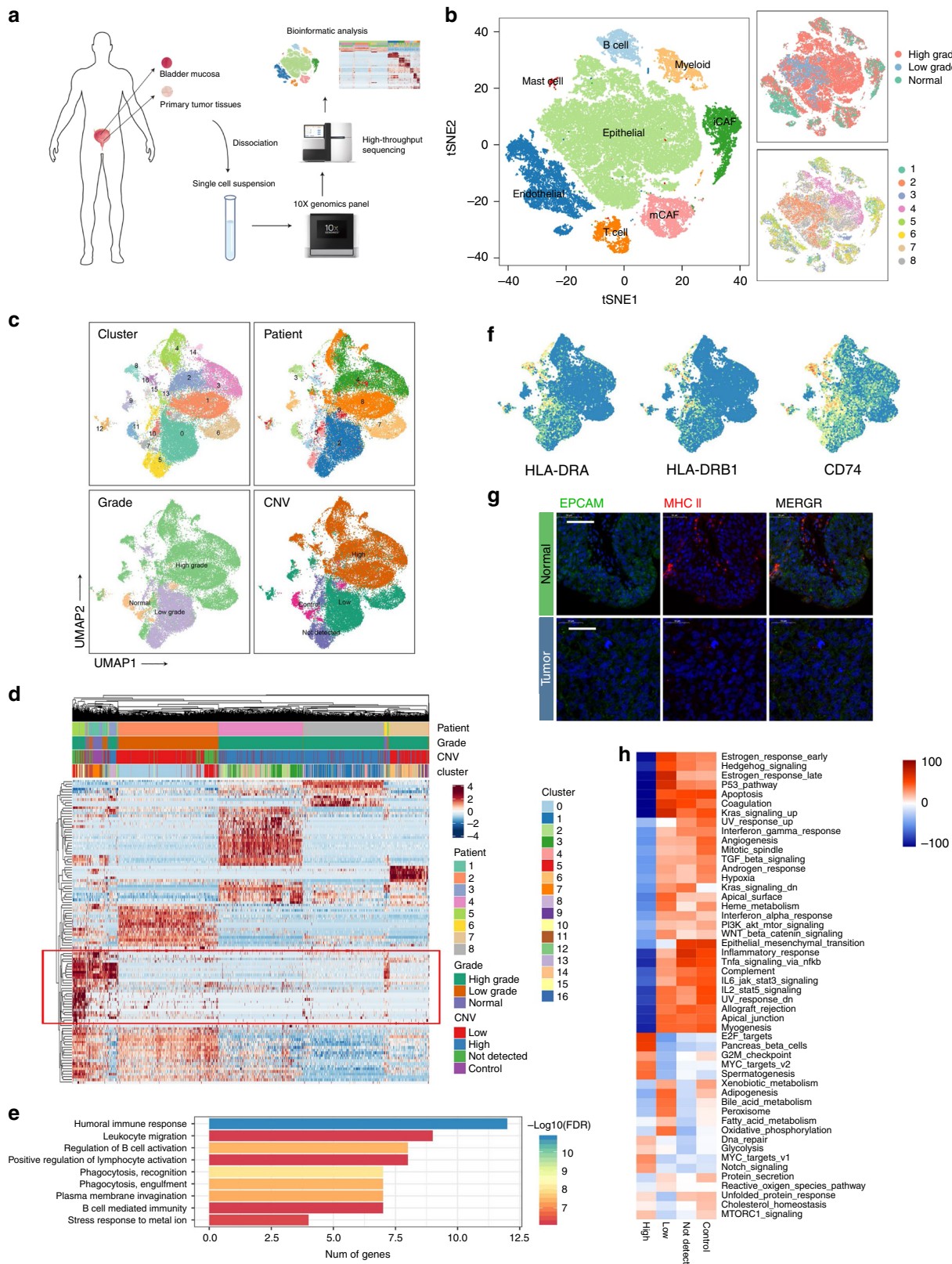

**Fig. 1 Identifying infiltrated cell types in BC and non-malignant tissues. a**, **b** Identifying infiltrated cell types in BC and non-malignant tissues. **a** Workflow of the sample preparation, sequencing and bioinformatic analysis. **b** tSNE plot of single cells profiled in the presenting work colored by major cell types, tumor grade and patient. **c**–**h** Reclustering of EPCAM+ cells. **c** UMAP plot EPCAM+ cells (epithelial marker) colored by cluster, patient, grade and CNV level. **d** Heatmap of differentially expressed genes (DEGs) of every CNV group. **e** Enriched GO functions of downregulated genes in malignant cells. **f** Expression levels of MHC-II molecules and CD74. **g** Immunofluorescence (IF) staining of MHC-II molecules and EPCAM. Scale bar represents 50 µm. **h** Heatmap shows difference in pathway activities scored by GSVA per cell between different CNV groups. Shown are *t*-values from a lineal model.

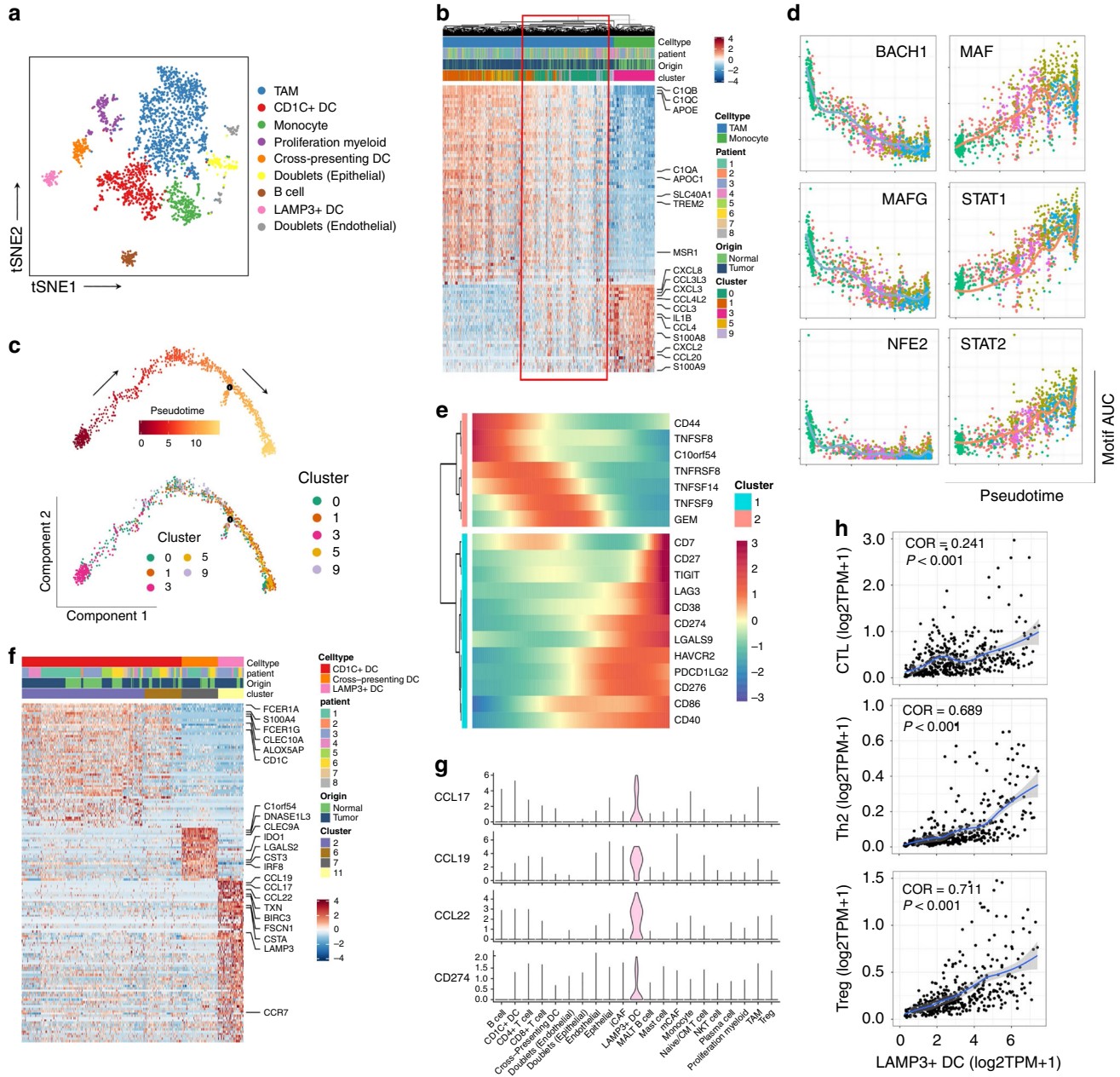

**Fig. 2 Reclustering of myeloid-derived cells (LYZ+). a** tSNE plot of subgroups of LYZ+ single cells. **b** DEGs between monocytes and TAMs. Single cells in red blank show features of both groups. **c** Trajectory of differentiation from monocyte into TAMs predicted by monocle 2. **d** Significantly inhibited or activated TF motifs in the differentiation process colored by cell clusters. **e** Heatmap show upregulated or downregulated immune checkpoints in the differentiation process. **f** Heatmap of DEGs between three different DC subgroups. **g** Violin plot show expression level of cytokines and CD274 highly expressed in LAMP3+ DCs. **h** Correlation between LAMP3+ DCs and different T cell subgroups in TCGA BLCA cohort. Coefficient was calculated with spearman correlation analysis.

LAMP3+ DCs showed the highest level of CD274 (Fig. 2f), which was even higher than what was observed in Tregs from BC tissues, indicating that this DC subgroup could inhibit CD8+ T cells directly or via recruiting Tregs into the tumor region. SCENIC analysis revealed that RELB and KDM2B motifs were highly activated in LAMP3+ DCs (Supplementary Fig. 5E–G).

**Two different fibroblast subtypes are identified in BC.** Fibroblasts (COL1A1+) were clustered into two different types: PDGFRA+ fibroblasts exhibit strong expression of various cytokines and chemokines, including CXCL12, IL6, CXCL14,

CXCL1, and CXCL2, which is similar to iCAFs described by Öhlund D et al. in a pancreatic cancer model[12]; RGS5+ fibroblasts have characteristics that are similar to myo-cancer-associated fibroblasts (mCAFs) (Fig. 3a, b). Existing iCAFs and mCAFs were assessed in tumor and non-malignant bladder stroma tissue by immunofluorescence (Fig. 3c). The results demonstrated that CAFs possess analogous subgroups across cancer types[12,13].

To investigate the function of each subgroup, we performed GO enrichment analysis on the DEGs of iCAFs and mCAFs. As shown in Fig. 3d, iCAFs were related to extracellular matrix organization, regulation of cell migration, and angiogenesis,

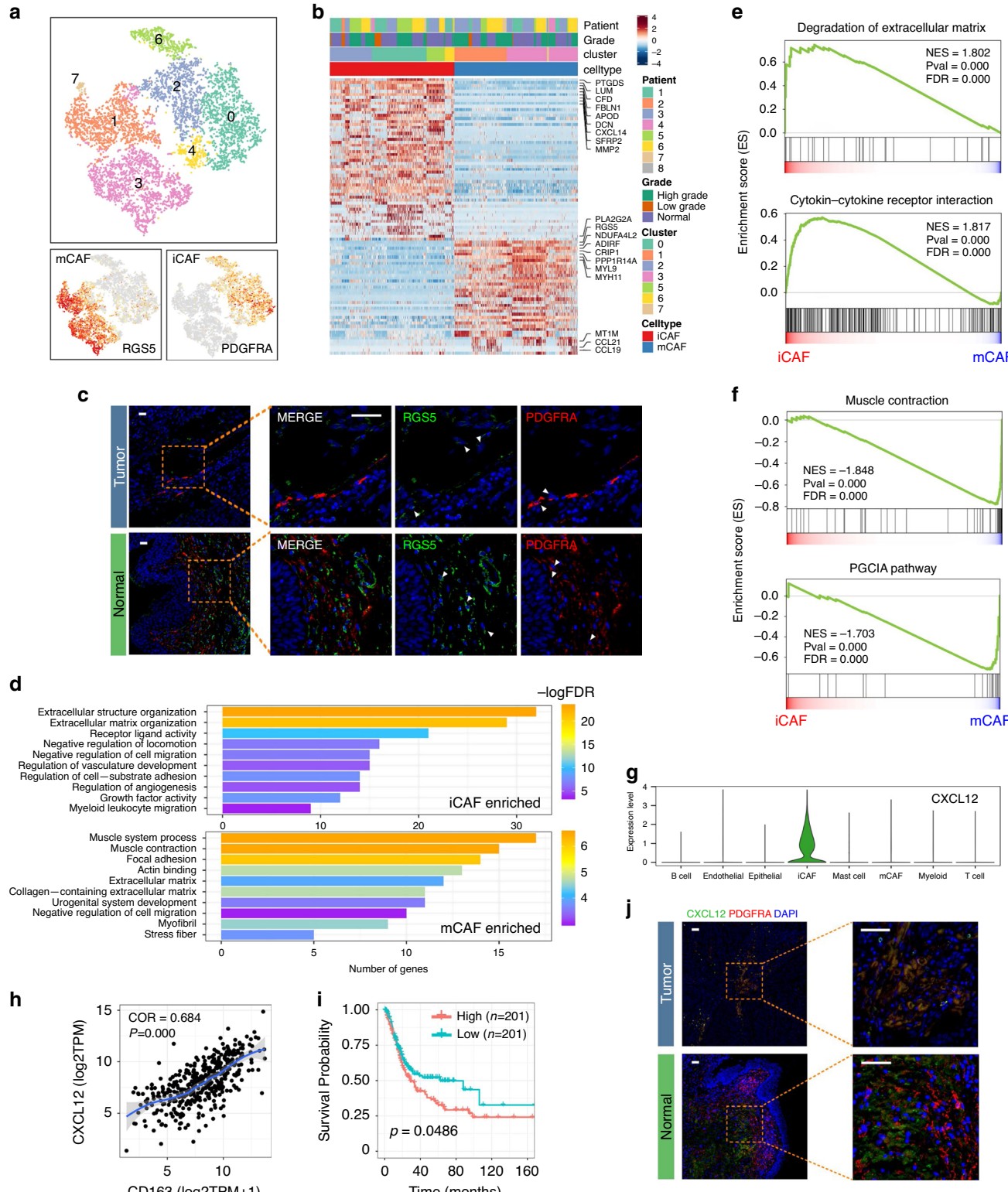

**Fig. 3 Fibroblasts in BC could be divided into two different subgroups. a** tSNE plot of fibroblasts colored by clusters (up) and subgroup markers (down). **b** Heatmap of DEGs between different fibroblast subgroups. **c** IF confirmed the existence of iCAFs and mCAFs ($n = 30$). Scale bar represents 50 μm. **d** Enriched GO functions of upregulated genes in iCAFs and mCAFs. **e, f** GSEA shows top enriched pathways in iCAFs (3E) and mCAFs (3F). NES denotes normalized enrichment score. **g** Violin plot shows expression level of CXCL12 across major cell types. **h** Correlation between CXCL12 level and tumor-infiltrated macrophages in TCGA BLCA cohort. Coefficient was calculated with spearman correlation analysis. **i** High level of CXCL12 predicted poor prognosis in TCGA BLCA cohort. Log-rank p value < 0.05 was considered as statistically significant. **j** IF recognized CXCL12+ iCAFs in BC tissues. iCAFs are the major derivation of CXCL12 in tumor tissues. Scale bar represents 50 μm.

whereas the muscle system process, focal adhesion, and extracellular matrix-associated pathways were significantly enriched in mCAFs. GSEA similarly revealed that iCAFs were associated with extracellular matrix degradation, indicating a potential role in tumor metastasis. The cytokine–cytokine receptor interaction pathway was also enriched in iCAFs. In contrast, muscle contraction and the PGC1A pathway were enriched in mCAFs, corresponding to a previous in vitro[12] study (Fig. 3e, f).

Since the cytokine–cytokine receptor interaction was enriched in iCAFs, we investigated the expression level of cytokines in the BC TME. Dramatically, iCAF was the major source of CXCL12, which is related to the accumulation of TAMs via CXCL12/CXCR4 interactions[14]. Notably, CXCL12 was positively correlated with the TAM signature in the TCGA BLCA cohort. A higher level of CXCL12 was significantly associated with a poor prognosis. Immunofluorescence staining confirmed that CXCL12 was expressed by iCAFs in BC tissues (Fig. 3g–j).

Via SCENIC analysis, we identified essential motifs in both CAF subgroups. MEF2D and MEF2C are mCAF-specific motifs that have profound roles in the transcriptional regulation of muscle lineages[15]. TCF21 and TWIST2 motifs were highly activated in iCAFs (Fig. 4a, b). In a previous study, TCF21 was found to be associated with coronary heart disease, enhancing the "fibromyocyte" phenotype of smooth muscle cells[16]. TWIST2 is a driver of epithelial–mechanism transition (EMT). However, their roles in CAF are still unknown.

### iCAFs have a pro-proliferation effect in BC.
iCAFs were predicted to have growth factor activity by GO enrichment. To confirm this hypothesis, we analyzed the expression levels of VEGF, FGF and IGF families in the BC TME (Fig. 4c). Dramatically, iCAFs are the major source of various growth factors. Among these growth factors, IGF1, an iCAF-specific group factor, was related to poor overall survival (Fig. 4d, e).

To validate its pro-proliferation effect, we sorted iCAFs by flow cytometry and co-cultured them with bladder urothelial carcinoma cell lines in vitro. Significantly, the co-culture group showed higher proliferation ability, which confirmed the protumor role of iCAFs in BC (Fig. 4f, g).

### Correlating scRNA-seq to public datasets.
To investigate the clinical role of the cell types identified in the present study, we evaluated the fraction of every cell type in samples from the TCGA BLCA cohort with CIBERSORTx[17] (Supplementary Fig. 7A–C, see "Methods"). Strikingly, only the accumulation of iCAFs and mCAFs was related to poorer overall survival (OS) (Fig. 5a, b). When correlating the cell fraction data with clinical information, we noticed that the cell type abundance was altered greatly across the molecular subtypes described in a previous study[4] (Fig. 5c, d). Luminal papillary, with the highest tumor purity, had the best prognosis, while the OS of the other four groups did not show a significant difference. Basal squamous showed the lowest tumor purity, as almost all cell types identified in our work occurred in this group. Notably, T cells were also enriched in this group, indicating that anti-immune checkpoint therapy may be suitable for these patients. In contrast to other cell types, mCAFs and iCAFs were enriched in four groups, with the exception of the luminal-papillary group. Since the luminal-papillary group contained mostly early stage samples, these results demonstrated that CAFs were closely related to the tumor progression of BC. In addition, we noticed that fibroblast markers were significantly downregulated in tumor tissues, while epithelial markers were upregulated. This could explain why downregulated DEGs were enriched in the extracellular region[18]. Since normal

bladder mucosa mainly contains EPCs, while fibroblasts are mainly located in stromal tissues, this phenomenon is probably a result of sampling depth. This could obscure the identification of real DEGs between normal epithelial and tumor cells (Supplementary Fig. 7D, E).

Subsequently, we expanded the clinical cohort, collecting over 3000 microarray profiled BC and non-malignant mucosa samples from the GEO and ArrayExpress databases (see "Methods"). A possible batch effect was eliminated with Combat function, and then 2959 tumor samples were clustered into five major groups by ConsensusClusterPlus[19] (Supplementary Fig. 8A–C). Four major clusters revealed similar characteristics to the luminal-papillary, luminal, luminal-infiltrated, and basal-squamous clusters. Another cluster, which was unlike Neural group in TCGA BLCA, showed both luminal-squamous and Luminal characteristics, so it was named as Luminal transition. As shown in Fig. 5e–h, OS in this meta cohort highly corresponded to that of TCGA, demonstrating the important role of CAFs in BC progression. iCAF and mCAF have many similar features, which may make the result of CIBERSORTx insufficient. In the cohort from TCGA, the iCAF-specific marker PDGFRA was significantly related to poor OS in BC patients, while the mCAF marker RGS5 was not, indicating that iCAF may have a more important role than mCAF (Supplementary Fig. 7F).

### Constructing an iCAF-based regulatory network for BC.
Using CellphoneDB2[20], we investigated the cell–cell interaction network among the cell types identified in our present work. Notably, iCAFs showed the most interactions with other cell types, and they showed especially strong interactions with ECs (Fig. 6a). Considering the results of GO analysis and GSEA and the expression abundance in our data, we collected data on interaction pairs including growth factors, CCL and CXCL families (Supplementary Fig. 9A).

iCAFs express higher levels of CXCL12, the receptors of which includes DPP4, CXCR3, CXCR4, and ACKR3 (CXCR7). As CXCR4 and CXCR3 are widely expressed on immune cells, secretion of CXCL12 by iCAFs is responsible for the immune infiltration status of BC. CCL2 and CXCL1 secreted by iCAFs could interact with ACKR1, which is highly expressed on the surface of ECs. These cytokines were previously reported to be associated with metastasis of BC[21,22], and we pinpointed their origin to iCAFs (Fig. 6b, d).

We also suggested that iCAFs produce VEGF, including VEGFA and VEGFB, which bind to VEGF receptors (FLT1, KDR, MET, and FLT4) on ECs to promote angiogenesis. In addition, FGFR1 was expressed on iCAFs and ECs, while FGFR3 was expressed on tumor cells. These receptors could bind to FGF, including FGF2 and FGF7 derived from iCAFs, and show pro-proliferation effects. IGF1R, a receptor for IGF1, was expressed on tumor cells and stroma cells, which suggested that iCAF could also factor into resistance to cisplatin[23]. Notably, tumor cells express high levels of VEGFA, which has the strongest proangiogenic ability. In addition, high-grade tumor cells highly express IGF2, interact with IGF2R on iCAFs, and promote tumor progression[24] (Fig. 6c, e). Together, our results predicted that iCAFs could promote the proliferation of tumor cells and stromal cells and potentially be able to recruit immune cells into the tumor stage.

### Discussion
Treatment of bladder urothelial carcinoma, especially muscle invasive bladder urothelial carcinoma is still a big problem until now. Anti-immune checkpoint therapy, including PD1/PD-L1, benefits only nearly 30% of patients with advanced disease.

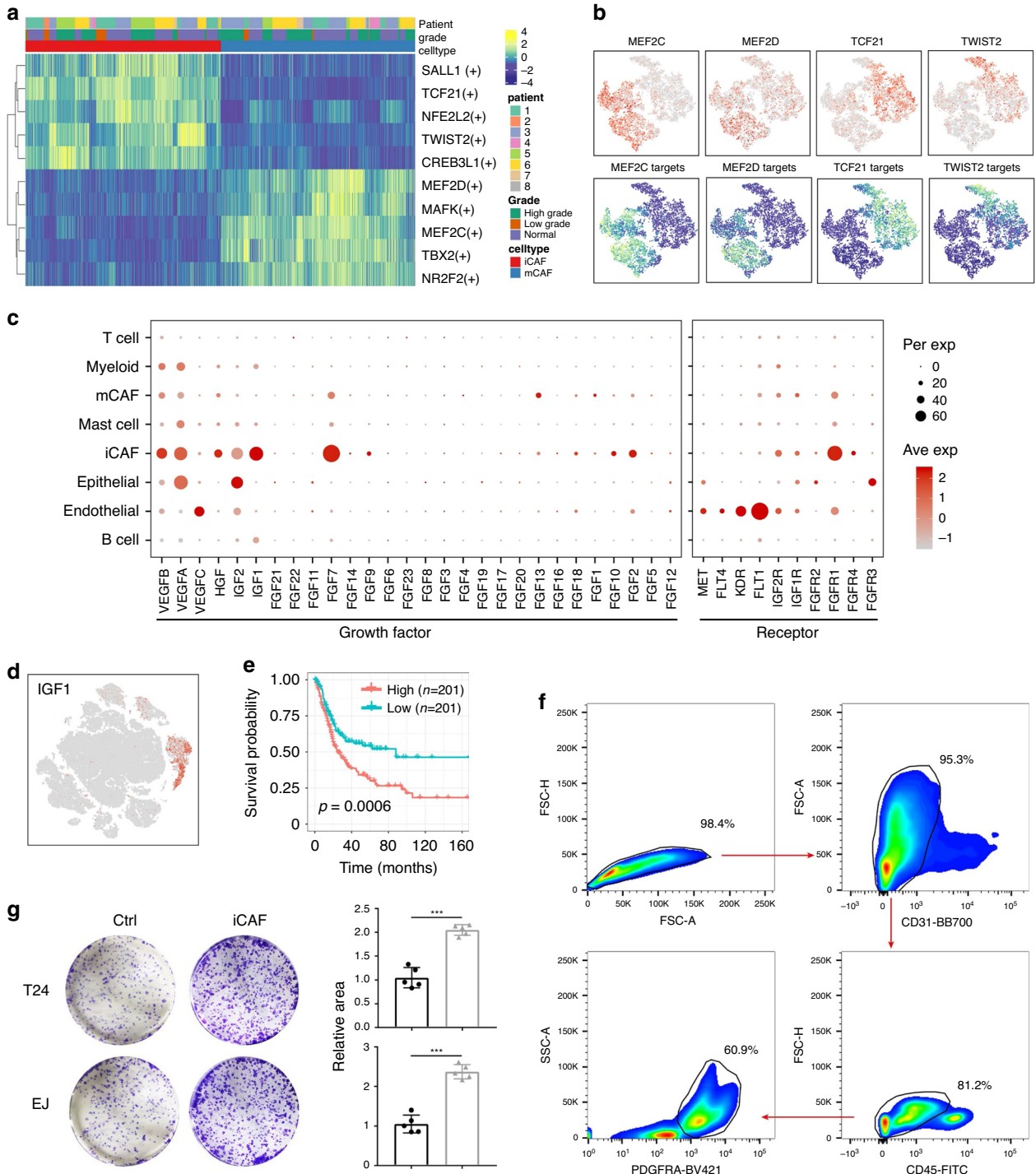

**Fig. 4 iCAFs promote proliferation of cancer cells. a** Heatmap of the area under the curve (AUC) scores of TF motifs estimated per cell by SCENIC. Shown are top five differentially activated motifs in iCAFs and mCAFs, respectively. **b** tSNE plots of the expression levels of TFs (up) and AUC scores (down). **c** Dot plot shows the expression level of growth factors across cell types. iCAFs are the major producer of growth factors. **d** tSNE plot shown the expression level of IGF1. IGF1 is secreted almost only by iCAFs. **e** High level IGF1 represents poor overall survival in TCGA BLCA cohort. *P* value was calculated with log-rank test. **f** FACS sorting strategy of iCAFs. **g** Co-culture and colony formation experiment showed that iCAFs have pro-proliferation property in vitro (*n* = 5). Error bar: mean value ± sd. *P* values were determined by two-side Student's *t* test. ***$p < 0.001$.

New targets or combined therapy strategies are still waiting to be discovered, and such discoveries could be hastened by a better understanding of the TME of bladder urothelial carcinoma. Here, we generated a single-cell transcriptome atlas and revealed the components of the microenvironment inside BC tissues.

In the present study, we identified 19 different cell types in the BC microenvironment. We suggest that the downregulated immunogenicity of cancer cells potentially contributes to the formation of an immunosuppressive microenvironment. We found that LAMP3+ DCs were related to the recruitment of Tregs and other CCR4+ immune cells. Since blocking of CCR4

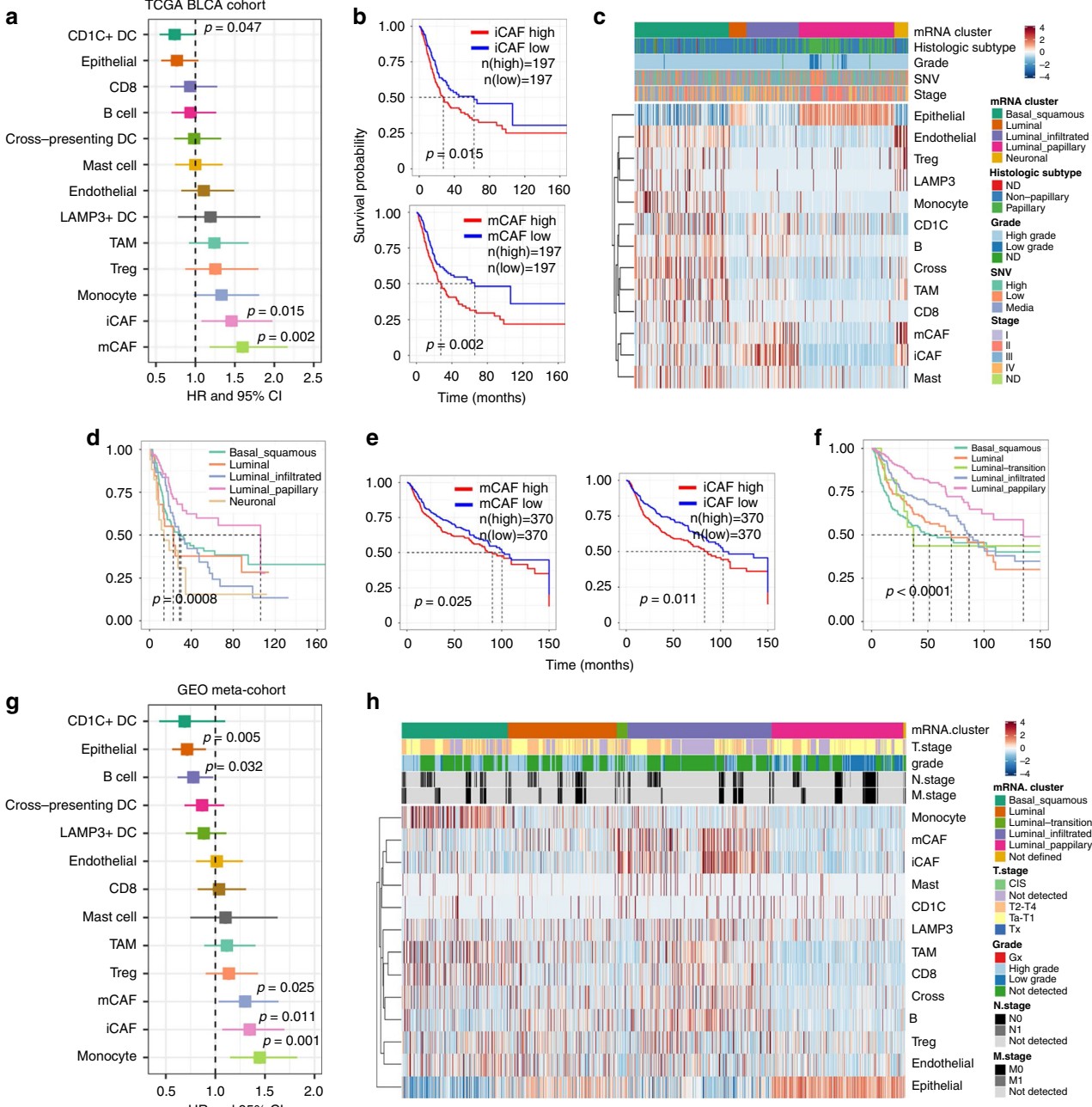

**Fig. 5 Molecular subtypes of BC were caused by heterogeneity of TME. a** Association between relative cell abundance and patient survival from TCGA BLCA cohort (COX regression analysis). **b** Kaplan–Meier curves for TCGA BLCA patients. *P* value was calculated with log-rank test. **c** Heatmap of cell abundance predicted per sample from TCGA BLCA cohort by CIBERSORTx. Shown are row *z*-score. **d** Kaplan–Meier survival curve for TCGA BLCA patients, grouped by molecular subtypes. **e** Kaplan–Meier survival curves for microarray-based meta-cohort patients. **f** Kaplan–Meier survival curve for microarray-based meta-cohort patients, grouped by molecular subtypes. *P* values of (**d**–**f**) were calculated with log-rank test. **g** Association between relative cell abundance and patient survival from microarray-based meta-cohort (COX regression). **h** Heatmap of cell abundance predicted per sample from microarray-based meta-cohort by CIBERSORTx. Shown are row *z*-score.

could significantly reduce Tregs recruited in the tumor stage in a canine model, LAMP3+ DCs may also be a potential target for immune therapy[25]. Furthermore, we also suggested that stromal cells, especially iCAFs, have strong pro-proliferation properties, which has rarely been discussed in BC models. We also produced a regulatory network based on iCAFs in a bioinformatic way. These results will help elucidate the role of stomal cells in BC. In the future, more function assay may help further understand the underline mechanism.

By correlating our results with those from public databases, we demonstrated that CAFs, especially one subset of them, iCAFs, were the key factor in tumor progression in BC. At present, immune therapy mainly targets T cells (PD1/PD-L1, CTLA-4)[26] or TAMs (CSF1R)[27]. However, these immune cells almost do not exist in a substantial number of BC patients, which may cause there to be no response for such patients to immune checkpoint inhibition therapy. Since CAFs were detected in almost all BC patients in an advanced stage, and since they were the major

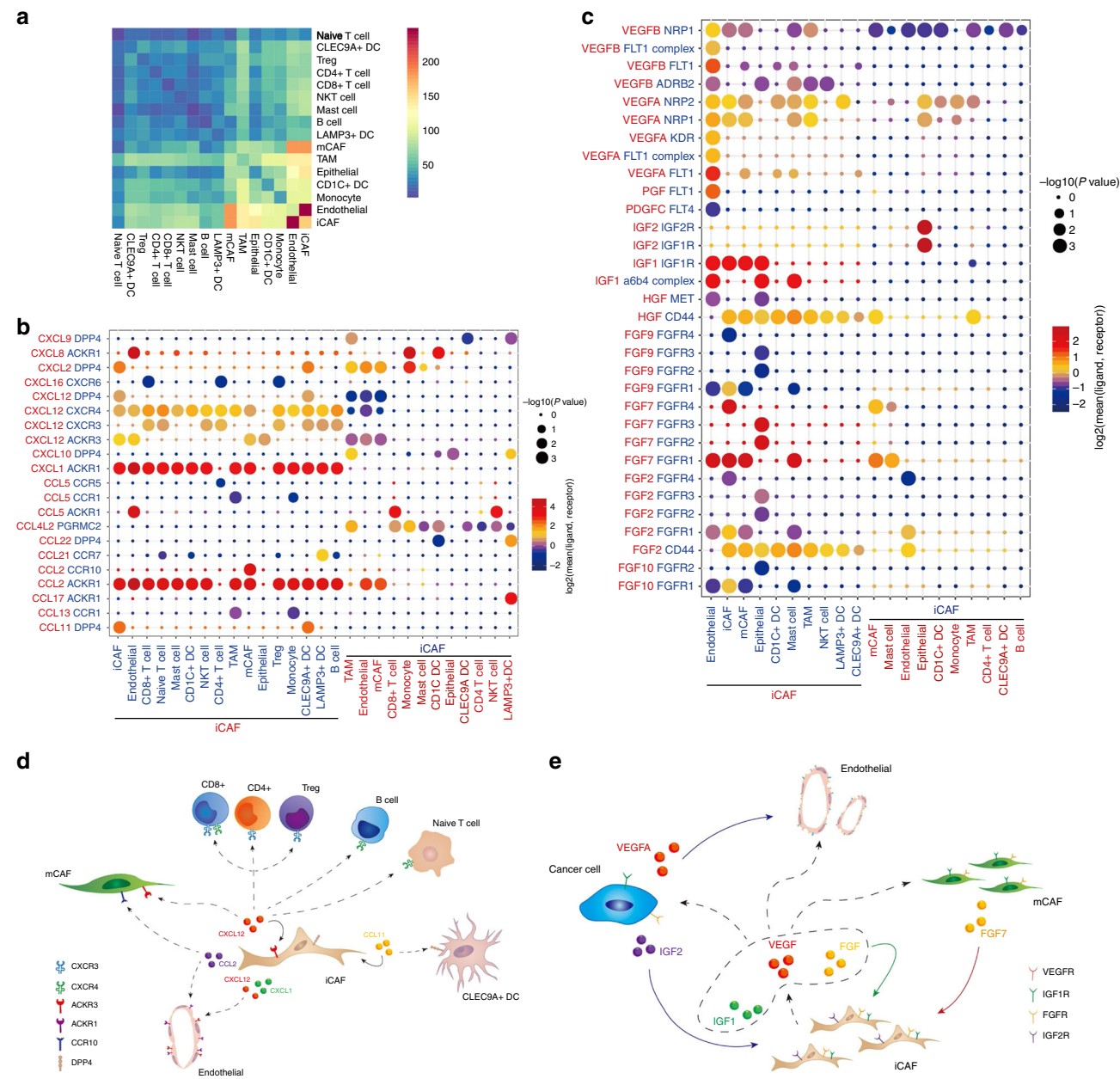

**Fig. 6 Cell–cell communication network in BC TME. a** Heatmap show number of potential ligand-receptor pairs between cell groups predicted by CellphoneDB 2. **b, c** Bubble plots show ligand-receptor pairs of cytokines **b** and growth factors **c** between iCAFs and other cell groups. **d, e** Predicted regulatory network centered on iCAFs.

source of various validated protumor growth factors in the TME, targeting CAFs may be an optimal choice for BC treatment.

Although we highlighted the role of iCAFs in this pipeline instead of mCAFs, the role of mCAFs in progression of cancer still could not be excluded. Limited by the deconvoluting algorithm, it's still difficult to identify accurate proportion of iCAFs and mCAFs in bulk sequencing data, which have highly similar features in transcriptome. More proofs are needed to be validate the role of mCAFs in BC.

Since fresh samples are needed in current single-cell sequencing strategy, batch effect could be involved between samples loaded in batches, which is still a big problem in bioinformatic analysis of single-cell sequencing data. Previously, Tran et al. has systemically compared the efficacy of the most prevalent tools for eliminating batch effects of single-cell profiling data[28]. Hence, Seurat V3 and Harmony are the most efficient tools. Since

then, batch derived difference was removed with Seurat V3 in this study.

In summary, we identified the expression profiles of subsets of cell in bladder urothelial carcinoma and confirmed the characteristics of these tumor-associated subsets. This cell atlas provides deep insight into cancer immunology and is an essential resource for drug discovery in the future.

## Methods

**Patients and samples.** All samples were obtained from the Union Hospital of Tongji Medical College, Huazhong University of Science and Technology, Wuhan, China. Eight primary bladder tumor tissues (2 low-grade bladder urothelial tumors, 6 high-grade bladder urothelial tumors) along with 3 adjacent normal mucosae, were involved in this cohort. Patients provided informed consent for this work. All experimental procedures were approved by the Institutional Review Board of Tongji Medical College, Huazhong University of Science and Technology (IRB: S116).

**Single-cell suspension preparation**. Bladder tumors and adjacent normal mucosa were processed immediately after being obtained from bladder cancer patients. Every sample was cut into small pieces (<1 mm in diameter) and then was incubated with 2 ml of trypsin (Gibco, Cat: R001100), 1 ml of collagenase II (Biofrox, Cat: 2275MG100) and 100 μl of DNase (Servicebio, Cat: 1121MG010) for 1 h on a 37 °C shaker. Subsequently, 4 ml DMEM was added to dilute the suspension, and then a 40-μm cell mesh was used to filter the suspension. After centrifugation at 250g for 5 min, the supernatant was discarded, and then the cells were washed with PBS twice. Then, the cell pellet was resuspended in 1 mL of ice cold red blood cell lysis buffer and were incubated at 4 °C for 10 min. Next, 10 ml of ice-cold PBS was added to the tube, and it was then centrifuged at 250g for 10 min. After decanting the supernatant, the pellet was resuspended in 5 ml of calcium- and magnesium-free PBS containing 0.04% weight/volume BSA. Finally, 10 μl of suspension was counted under an inverted microscope with a hemocytometer. Trypan blue was used to quantify live cells.

**Droplet-based single-cell sequencing**. According to the manufacturer's protocol, Chromium Single cell 3′ Reagent v3 kits were used to prepare barcoded scRNA-seq libraries. Single-cell suspensions were loaded onto a Chromium Single-Cell Controller Instrument (10 × Genomics) to generate single-cell gel beads in emulsions (GEMs). To capture 8000 cells per library, approximately 12,000 cells were added to each channel. After generation of GEMs, reverse transcription reactions were engaged to generate barcoded full-length cDNA, which was followed by disruption of emulsions using the recovery agent, and then cDNA clean-up was performed with DynaBeads Myone Silane Beads (Thermo Fisher Scientific). Next, cDNA was amplified by PCR for the appropriate number of cycles, which depended on the number of recovered cells. Subsequently, the amplified cDNA was fragmented, end-repaired, A-tailed, and ligated to an index adaptor, and then the library was amplified. Every library was sequenced on a HiSeq X Ten platform (Illumina), and 150 bp paired-end reads were generated.

**Raw data processing and quality control**. Cell Ranger (version 2.2.0) was used to process the raw data, demultiplex cellular barcodes, map reads to the transcriptome, and down-sample reads (as required to generate normalized aggregate data across samples). These process produced a raw unique molecular identifier (UMI) count matrix, which was converted into a Seurat object by the R package Seurat[29] (version 3.0.0). Cells with UMI numbers <1000 or with over 10% mitochondrial-derived UMI counts were considered low-quality cells and were removed. In order to eliminate potential doublets, single cells with over 6000 genes detected were also filtered out. Finally, 52721 single cells remained, and they were applied in downstream analyses.

After quality control, the UMI count matrix was log normalized. Since sample from eight patients were processed and sequenced in batches, patient number was used to remove potential batch effect. In this process, top 3000 variable genes were used to create potential Anchors with FindIntegrationAnchors function of Seurat. Subsequently, IntegrateData function was used to integrate data and create a new matrix with 3000 features, in which potential batch effect was regressed out.

To reduce the dimensionality of the scRNA-Seq dataset, principal component analysis (PCA) was performed on an integrated data matrix. With Elbowplot function of Seurat, top 30 PCs were used to perform the downstream analysis. The main cell clusters were identified with the FindClusters function offered by Seurat, with resolution set as default (res = 0.8). And then they were visualized with 2D tSNE or UMAP plots. Conventional markers described in a previous study were used to categorize every cell into a known biological cell type. Firstly, 52721 cells were clustered into seven major cell types. Subsequently, every major cell type was subset and further clustered into subclusters to detect heterogeneity within every cell type, respectively. The Seurat Findallmarker function was performed to identify preferentially expressed genes in clusters or differentially expressed genes between tumor- and normal-derived cells. Rds of total dataset, myeloid lineage and fibroblast have been uploaded as Supplementary Software 1, which could be read in R environment by Shiny R package.

**Estimation of CNVs in cancer cells**. The InferCNV package was used to detect the CNVs in EPCAM+ cells and to recognize real cancer cells with default parameters. Two clusters mainly containing non-malignant derived cells were used as the control group.

**Trajectory and RNA velocity analysis**. To map differentiation in the TME, pseudotime analysis was performed with Monocle2[8] to determine the dramatic translational relationships among cell types and clusters. Further detection with the Monocle2 plot_pseudotime_heatmap function revealed the key role of a series of genes in the differentiation progress. Signifcantly changed genes were identified by the differential GeneTest function in Monocle2 with a q-value < 0.01.

RNA velocity was performed to investigate potential inter-relationship of myeloid lineage. BAM file containing all the myeloid cells was used in this pipeline. All the parameters were set as default. he result was visualized into UMAP plot.

**Simultaneous gene regulatory network analysis**. CENIC[30] is a new computational method used in the construction of regulatory networks and in the

identification of different cell states from scRNA-seq data. To measure the difference between cell clusters based on transcription factors or their target genes, SCENIC was performed on all single cells, and the preferentially expressed regulons were calculated by the Limma package[31]. Only regulons significantly upregulated or downregulated in at least one cluster, with adj. p-value < 0.05, were involved in further analysis.

**Cell–cell communication analysis with CellPhoneDB 2**. CellPhoneDB 2 is a Python-based computational analysis tool developed by Roser Vento-Tormo et al [20], which enables analysis of cell–cell communication at the molecular level. A website version was also provided for analysis of a relatively small dataset (http://www.cellphonedb.org/). As described above, 52721 single cells that were clustered into 19 cell types were investigated using the software to determine interaction networks. Interaction pairs whose ligands belong to the VEGF, FGF, CCL, or CXCL families and have P-values < 0.05 returned by CellPhoneDB, were selected for the evaluation of relationships between cell types.

**Correlation to public datasets**. Transcriptome data from The Cancer Genome Atlas (TCGA) BLCA datasets were obtained from UCSC XENA (https://xena.ucsc.edu/). Clinical information was presented by Robertson et al.[4] in their supplementary materials.

For cell subgroups, genes with FC >2 were considered as marker genes, Mean TPM level of marker genes were log2 transformed and used as gene signature. Spearman correlation analysis was used to estimate correlation between specific cell types.

A microarray-based study of bladder urothelial carcinoma containing over 15 tumor samples in the GEO or ArrayExpress databases was downloaded and normalized into a meta-cohort. Batch effects between cohorts based on the same platform were initially removed by the Combat function of the sva package[32]. Subsequently, batch effects between different platforms were also removed. 3D PCA was performed to confirm the efficiency of Combat. ConsensuClusterPlus[19] was used to detect major molecular clusters in the meta-cohort.

To evaluate the relative abundance of each cell type identified in the present study, cibersortx[17] was performed with default parameters. Subsequently, the relative cell abundance was divided into high 50% and low 50%. Kaplan–Meier analysis was performed to evaluate the prognostic value of cell clusters and detect the role that these cell clusters play in bladder cancer progression. All these analyses were performed in R (3.6.0).

**Pathway analysis**. Differentially expressed genes (DEGs) of cell subgroups were recognized by the findmarker function provided by Seurat. |FC| > 2 and adj.p.val < 0.05 were used as the cut-off criteria. GO enrichment analysis was performed on these DEGs with clusterProfiler[33]. GSEA was performed on a matrix of all genes detected by the desktop tool downloaded from http://software.broadinstitute.org/gsea/index.jsp. GSVA was conducted with the GSVA package[34]. Differences between different cell groups were calculated with a linear model offered by the Limma package.

**Immunofluorescence staining**. A tissue chip, HBla-U060CS-01, which consists of 30 paired tumor and paratumor samples, was purchased from Shanghai Outdo Biotech. To ensure the consistency of the analysis, all immunofluorescence analyses were performed using the same type of tissue chip.

The following antibodies were used to detect specific proteins: anti-HLA-DRA (rabbit, 1:50, Proteintech, Cat. No. 11221-1-AP), anti-EPCAM (rabbit, 1:50, Proteintech, Cat. No. 21050-1-AP), anti-RGS5 (rabbit, 1:50, Proteintech, Cat. No. 11590-1-AP), anti-PDGFRA (rabbit, 1:200, Abcam, ab203491), and anti-CXCL12 (rabbit, 1:100, Abcam, ab155090).

**Flow cytometry experiments**. Tissue samples were disassociated as described above. CD31-CD45lowPDGFRA+ iCAFs were collected by flow cytometry. The following antibodies were used: anti-CD31 (mouse, 1:300, BD, 566563), anti-CD45 (mouse, 1:100, BD, 555482), and anti-PDGFRA (mouse, 1:500, BD, 562799). Anti-CD45 (mouse, 1:100, BD, 555482), anti-CD3 (BD, 1:200, 555335), anti-CD14 (BD, 1:200, 563079), anti-CCR7 (BD, 1:200, 557734), anti-CLEC9A (BD, 1:200, 564266) and anti-CD1c (BD, 1:200, 564900) were used to confirm the subgroups of myeloid lineage. Data analysis was performed in FlowJo (V10).

**Co-culture colony formation experiments**. T24 and EJ bladder urothelial carcinoma cell lines were obtained from American Type Culture Collection (ATCC) and were cultured according to standard protocols. A co-culture experiment was performed by seeding bladder cancer cells ($2 \times 10^3$) in the lower chamber and iCAFs ($1 \times 10^5$) in the upper chamber of a 6-well transwell apparatus with a 0.4 μm pore size (Corning Incorporated, NY, USA), and the cells were cultured together for 7 days. Subsequently, colonies were fixed with 4% paraformaldehyde and then were stained with crystal violet. The areas of the colonies were estimated by ImageJ. Student's t tests were used to detect differences between the control and co-culture groups.

**Statistical analysis**. All statistical analyses and graph generation were performed in R (version 3.6.0) and GraphPad Prism (version 7.0).

**Reporting summary**. Further information on research design is available in the Nature Research Reporting Summary linked to this article.

## Data availability

The single-cell RNA sequencing data generated in this paper is available in GSA-Human under the accession code HRA000212 and in SRA datasets under BioProject PRJNA662018. Affymetrix microarray datasets including GSE83586[35], GSE8730[36], GSE31684[37], GSE104922[38], GSE124305[39], GSE39016[40], GSE71576[41], GSE57933[42], GSE38264[43], GSE31189[44], GSE3167[45], GSE3731[46], GSE5287[47] from GEO and E-MTAB-1803[48], E-MTAB-1940[49] from ArrayExpress, along with Illumina microarray datasets, including GSE48075[50], GSE32894[51], GSE1350[52], GSE48276[50], GSE6979[53], GSE70691[53], GSE120736[54], GSE8641[55], GSE57813[56], GSE3254[57], GSE52219[50], GSE52329[50] were combined to construct the meta-cohort dataset. TCGA BLCA datasets from UCSC XENA (http://xena.ucsc.edu/) were also used in this study. All remaining relevant data are available in the article, supplementary information, or from the corresponding author upon reasonable request.

## Code availability

R scripts to read in R environment by Shiny R package, the Rds files of total dataset, myeloid lineage and fibroblast have been uploaded as Supplementary Software 1 as part of the Supplementary Information associated with this article. Other R scripts used to analyze data and generate figures are available upon request to the corresponding author.

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

## Acknowledgements

The authors thank OE Biotech Co., Ltd (Shanghai, China) for providing single-cell RNA-seq and Dr. Yongbing Ba and Yao Lu for assistance with bioinformatics analysis. We thank Bo Zhang from NovelBioinformatics Ltd., Co. for the support of bioinformatics analysis with their NovelBrain Cloud Analysis Platform (www.novelbrain.com). We also thank Dr. Yichen Lei of Ruijin Hospital for help of flow cytometry. We also thank Dr. Jianming Zeng(University of Macau), and all the members of his bioinformatics team, biotrainee, for generously sharing their experience and codes. This work was supported by grant from National Natural Sciences Foundation of China (81672524), Hubei Provincial Natural Science Foundation of China (2018CFA038), Science, Technology and Innovation Commission of Shenzhen Municipality (JCYJ20180305164838833) and Fundamental Research Funds for the Central University (201kfyRCPY0049).

## Author contributions

K.C. and Z.C. conceived the idea. L.Z. collected the specimen and prepared single-cell suspension for sequencing. J.H. finished the bioinformatics analysis. J.H., L.L., Y.H. finished the FACS sorting and co-culture assay. M.X., Y.Y. helped with immuno-fluorescence staining, J.H. and K.C. wrote the manuscript. All authors reviewed and approved the manuscript.

## Competing interests

The authors declare no competing interests.
