## [Peer Review File · Nature Communications]

Reviewers' comments:

Reviewer #1 (Remarks to the Author):

The authors have provided in depth look into bladder cancer using 10x genomics. Like any new tool, it provides a wealth of information that is overwhelming. Such large studies as this with over 52000 cells depends on statistical analysis with biological interpretation. With such large numbers, it's easy to obtain good statistical results though they may not have any biological significance. There is major dilemma regarding this manuscript: This manuscript provides a wealth of information that could potentially lead to new ideas/directions in the field. On the other hand, the manuscript that provides interesting data, unfortunately, does not contain substance.

Major comments:

- 1) The authors describe 19 cell types in BC tissues. Unfortunately, authors could not to provide the additional supporting evidence that validate the data analysis. BC patients are heterogenous and more so when they receive therapy. No matter how extensive the data analysis, the results should be validated in some way (flow cytometry or Western blotting or qRT-PCR). Ideally using the same samples and other BC samples with similar characteristics.
- 2) The additional information need to be provided for patients: age, sex, type of surgery (TURBT or full cystectomy), grade, therapies, tumor size.
- 3) Authors describe a total of 52721 cells from 8 patients that underwent 10x/RNA-seq. The methods state 8K to 12K cells per sample. With 8 samples, there appears to be large discrepancy regarding cell number. The amount of cells loaded on 10x can be complicated and lead loading variations. The author needs to provide details of cell numbers per each sample analyzed. If there is a large number of variation of cells from one sample to another, how is this accounted for? What would that mean for the analysis and interpretation of the results.
- 4) With such a large shotgun approach screening, some of the data should be validated either if not both RNA expression and protein level. In Fig 3c confirmation of iCAFs, how many patients were screened? Is this 1 out of the 8 patients? To be properly validated, iCAFs should be validated in BC patients outside of the 8 used. Similar reasoning for Fig 3D. What percentage of these cells are in the tumor? Are they located throughout or localized to certain portions?
- 5) With such patient variability even after normalizing for batch effect, number of 8 patients is too small of a sample size to obtain biological significance. It's not surprising to find statistical significance with a data set screen of 52721 cells considering each cell has 46000 protein encoding genes. It's not clear how p-values have been adjusted. Nor is it clear how such a small heterogeneous patient cohort can be interpreted. To be clear, the reviewer does not dispute the validity of the data in the paper, but is concerned that statistical results do not necessarily reflect biological relevance.

Reviewer #2 (Remarks to the Author):

To the Authors:

In this manuscript the authors used single-cell RNA-seq to profile the tumor microenvironment of bladder carcinoma. In total they analyzed 8 primary tumor samples and 3 para-tumor samples. In a first step they integrated all data and determined the different cell types within the microenvironment, which is followed by more specific analyses on individual populations using a variety of single-cell RNAseq tools and methods. They noticed that cancer heterogeneity between

patients is largely driven by copy number variations, which was directly inferred from the single-cell RNAseq data. Then, the authors apply trajectory analysis on myeloid cells and propose that monocytes undergo M2 polarization which might be the source for the observed increased tumor-associated macrophages (TAMs) in tumors. In addition, they also link dendritic cells (DC) to the recruitment of regulatory T-cells (Tregs). Finally, they focused on cancer associated fibroblasts and first divided them in myo-cancer-associated fibroblast (mCAFs) and inflammatory cancer associated fibroblasts (iCAFs). Hereby, they observed that iCAFs display increased expression of potential growth factors, which was correlated with increased growth potential of cancer cells in co-culture studies. Then, they used public datasets to assess the effect of such fibroblast populations and noted that they can be used as a negative predictor for overall survival. This is in line with their final model, where iCAFs are believed to promote tumor growth and potentially suppress immune cell populations by means of their (inferred) ligand-receptor signaling profile. While I believe that the dataset and conclusions could be very valuable for the cancer community, I have several concerns, which are necessary for the authors to address to make the presented paper and dataset useful to other researchers.

Major comments:

Point 1. The bulk of the paper is based on single-cell RNA-seq data and contains only few validation experiments. Nevertheless, the dataset here can be invaluable for the larger research community as it could aid in developing novel hypotheses or be used to cross validate with their own datasets. Therefore I have a number of specific comments or suggestions which are needed to improve the usability of the presented work:

1.1 The authors should refrain from using strong language when their results are based on correlative or associated conclusions and not validated by an additional experiment or technology. Hence, I would recommend the authors to be more prudent in their conclusions or alternatively provide additional experiments that could validate their novel findings.

Here are just a few examples:

- Lines 10-13: We identified 19 different cell types in the BC microenvironment, suggesting that the downregulated immunogenicity of cancer cells contributes to the formation of an immunosuppressive microenvironment.

It's a big step to link the presence of multiple cell types to an immunosuppressive microenvironment, especially since there is a large variability in cell type composition between different tumor samples.

- Lines 13-14: We also found that monocytes underwent M2 polarization in the tumor region and differentiated.

This statement is based purely on a trajectory analysis and for which – at least – additional analyses should be performed as further explained in point 2.4 below.

- Lines 15-16: Furthermore, the LAMP3+ DC subgroup, which has not been described in BC, recruited regulatory T cells, leading to the formation of an immunosuppressive TME.

This statement is based entirely on other literature and not proven in the manuscript.

I've limited the examples to the abstract only, but I would encourage the authors to carefully read the other parts of the paper as well.

1.2 Make the biological message of the paper straightforward and more clear by focusing more on one specific aspect. Ideally, this should be on the effect of the inflammatory cancer-associated fibroblasts on tumor cells, which they suggest is mediated through the inferred signaling pathways. A few simple experiments in line with Figure 4G could already answer several key questions:

Is the iCAF-tumor effect contact dependent or not? Is there a difference?

How important is one individual molecule (ligand or receptor) or is there redundancy between them?

1.3 Make it easier for others to access and explore the data. Provide scripts to reproduce the results and/or create a graphical user interface to browse through the data. The latter can be done relatively easy with an R Shiny app.

Point 2. The authors use a large repertoire of established computational methods to describe their obtained single-cell RNA-seq data, however details are majorly lacking. Since the obtained scRNA-seq results constitute the majority of the results the authors should do a significant effort to describe each method and decision in more detail. For example, which parameters did they use or test? Did they use data-driven or prior knowledge into their decision making? What was the rationale to use a specific method?

More specifically I have the following questions:

2.1 Since the authors use the 10X genomics platform to generate their single-cell RNAseq data it would be of interest to know which (or if) samples were processed in parallel? This is important since in their first step they use a method to integrate different datasets. They claim this is necessary to reduce 'batch' effects, however if (some) samples are indeed processed in parallel on this platform, then there should be only a minimal 'batch' effect and the authors might instead remove important biological variability. At minimum a rationale should be provided in the paper as to why they followed this approach.

2.2 Please provide which underlying commands and parameters in the Seurat pipeline have been used. Provide the default values if they have been used. Why did they decide on using 3000 variable genes and how did they determine the number of principal components to use downstream? How did they determine the resolution of the (graph-based Louvain?) clustering? When they subset a cell population, do they run the same exact pipeline again or is this different each time? Importantly, the command to integrate multiple datasets does not remove 'batch effects' per se. It identifies mutual nearest neighbors in lower dimensions to adjust the gene expression matrix, in other words it cannot discriminate between biological or technical variation in that process. As such I would suggest the authors to consider not to use the term 'batch effect' in this situation.

2.3 Which cells were exactly used as control group to infer copy number variations? These should also be epithelial (-like) cell types, as this type of analysis is based on gene expression changes and thus the control cell type could have a major influence on the obtained results.

2.4 What prior information was used to focus only monocytes and TAMs in the pseudotime trajectory analysis? The authors should apply RNA velocity (La Manno et al) on all the populations within the myeloid populations (Figure 2A) to identify their inter-relationships (and direction).

2.5 For the SCENIC analyses it would be useful to provide some supplemental figures illustrating the enriched regulons (TFs \diamond downstream genes).

2.6 Please explain how batch effects were removed with the sva package? Do all the arrays have common samples, which were then used to remove potential technical batch effects or how was this analysis performed? Is this really batch effect removal or "estimating surrogate variables for unknown sources of variation" (sva package)? Please, make sure to use the proper terminology.

Minor comments:

- Please provide technical details about the single-cell RNAseq quality (histograms, statistics, ..)
- Suppl. Fig. 2 and Fig. 1 Are any of the tumor and para-tumor samples matched? If so, did you

use that information in your analysis?

- Is IGF2 found on the common gained CNVs? Can the authors provide a list of genes that are on the common gained and deleted CNV regions?

- "Among the 3 DC subgroups, LAMP3+ DCs expressed various genes encoding 120 cytokines, including CCL17, CCL19 and CCL22 (Figure 2H)"

The figure call out is most likely wrong and should be Figure 2H.

- Figure 2H. Correlation is not an ideal estimator here. The correlation (r) result is likely dependent on the outliers as most of the datapoints are in the lower left part of the graph. Is this Pearson or Spearman? How are the signatures created?

- There is no Figure 4I and no Figure 4H

- Can the authors discuss why both iCAF and mCAF display a similar negative association with overall survival in Figure 5A? This does not seem to be in line with their other data.

- "Together, our results predicted that iCAFs could promote the proliferation of tumor cells and stromal cells and could also recruit immune cells into the tumor stage, leading to the establishment and maintenance of an immunosuppressive microenvironment."

This statement seems to be contradictory. The authors should explain better how increased recruitment of immune cells and an immunosuppressive microenvironment can co-exist. For example, increase in regulatory T-cells are also observed with strong immune activation, as this is believed to be a normal immunological feedback mechanism.

- As a follow-up on the previous comment: in supplemental Figure 2G the tumor samples display a large variability in cell-type composition, which does not seem to be associated with tumor grade. Can the authors comment or discuss what this might suggest in light of their model?

Reviewer #3 (Remarks to the Author):

The authors perform single cell analysis based on 8 tumors and 3 non-tumor samples from patients with bladder cancer. Overall, based on the number of tumor samples included, the claims in the paper are too strong. My overall concern is that if 8 new tumors are analyzed, they may show different patterns.

Some specific points:

1. Line 31: Not correct. Significant progress have been made in therapeutic strategies for bladder cancer patients in recent years.

2. Line: 80: I don't think the cancer cells are producing immunoglobulins – this claim seems strange

3. Line 281: patient samples, what kind of tumor stages are analyzed here? Outcome ? is it comparable to the MIBC TCGA data ?

4. In Figure 1C it looks like most of the iCAF cluster is driven primarily by one patient. In figure 3 this looks different – please explain.

5. Figure 1D – difficult to interpret this, what is actually shown in the different expression clusters (pathway enrichment etc). Overall, many of the figures (and the logistics behind) are difficult to follow. E.g. different cluster numbers are used (beginning from 0 and beginning from 1 – how should one follow this?). Also why are e.g. the specific clusters in 2c selected, and why are only two cluster selected in 2E.

Reviewer #1:

Major comments:

1) The authors describe 19 cell types in BC tissues. Unfortunately, authors could not to provide the additional supporting evidence that validate the data analysis. BC patients are heterogenous and more so when they receive therapy. No matter how extensive the data analysis, the results should be validated in some way (flow cytometry or Western blotting or qRT-PCR). Ideally using the same samples and other BC samples with similar characteristics.

Response: Thank for the reviewer's suggestion. We agree with the reviewer that the results obtained with scRNA sequencing should be validated in other way. In order to confirm the existence of two different fibroblast subgroups, immunofluorescence staining (IF) was performed on a tissue chip containing 30 paired tumor and normal bladder mucosa (**Figure 3C**). As shown in **Supplementary Figure 4A**, existence of monocyte, macrophage and 3 different dendritic cells are confirmed by flow cytometry in 5 BC tissues. Other immune cell subgroups in bladder carcinoma microenvironment, such as NK, Treg and Plasma cells have been reported in previous studies (Wang, T. *et al.*; Eckstein, M. *et al.*; Burke, B. *et al.*).

Reference:

Wang, T. *et al.* CCR8 blockade primes anti-tumor immunity through intratumoral regulatory T cells destabilization in muscle-invasive bladder cancer. *Cancer immunology, immunotherapy : CII* (2020).

Eckstein, M. *et al.* Cytotoxic T-cell-related gene expression signature predicts improved survival in muscle-invasive urothelial bladder cancer patients after radical cystectomy and adjuvant chemotherapy. *Journal for immunotherapy of cancer* **8** (2020).

Burke, B. *et al.* Inhibition of Histone Deacetylase (HDAC) Enhances Checkpoint Blockade Efficacy by Rendering Bladder Cancer Cells Visible for T Cell-Mediated Destruction. *Frontiers in oncology* **10**, 699 (2020).

2) The additional information need to be provided for patients: age, sex, type of surgery (TURBT or full cystectomy), grade, therapies, tumor size.

Response: The reviewer's suggestion has been well taken. The clinical information has been offered in **Supplementary Table 1**.

3) Authors describe a total of 52721 cells from 8 patients that underwent 10x/RNA-seq. The methods state 8K to 12K cells per sample. With 8 samples, there appears to be large discrepancy regarding cell number. The amount of cells loaded on 10x can be complicated and lead loading variations. The author needs to provide details of cell numbers per each sample analyzed. If there is a large number of variation of cells from one sample to another, how is this accounted for? What would that mean for the analysis and interpretation of the results.

Response: The reviewer requested that "The author needs to provide details of cell numbers per each sample analyzed". Thanks for raising this important question. Cell numbers per sample after quality control was shown in **Supplementary Figure 2B**.

The reviewer expressed their concern that the amount of cells loaded on 10x can be complicated and lead loading variations. As shown in **Supplementary Figure 2B**, although we loaded similar cell numbers per sample, there's still variation between samples. However, after clustering of single cells, none of the cell subgroups was clustered due to different sequencing depth (**Supplementary Figure 1B**), suggesting that variation of cell numbers per sample does not influence our conclusion. Similar phenomenon was observed in previous work of other team (Vento-Tormo, R., *et al.*).

Reference:

Vento-Tormo, R., *et al.* Single-cell reconstruction of the early maternal-fetal interface in humans. *Nature* **563**, 347-353 (2018).

4) With such a large shotgun approach screening, some of the data should be validated either if not both RNA expression and protein level. In Fig 3c confirmation of iCAFs, how many patients were screened? Is this 1 out of the 8 patients? To be properly validated, iCAFs should be validated in BC patients outside of the 8 used. Similar reasoning for Fig 3D. What percentage of these cells are in the tumor? Are they located throughout or localized to certain

portions?

Response: The reviewer suggested that the data should be validated either if not both RNA expression and protein level. Thanks for the important suggestion. As shown in Figure 1G, expression of EPCAM and MHC-II molecule were validated at protein level by immunofluorescence (IF). Existence of iCAFs was validated by IF staining (**Figure 3C**). Co-expression of iCAF marker (PDGFRA) and CXCL12 were confirmed in **Figure 3J**. iCAFs were also recognized by FACS sorting (**Figure 4F**). In addition, the co-culture assay revealed the pro-tumor role of iCAFs in bladder urothelial carcinoma (**Figure 4G**).

The reviewer asked “In Fig 3c confirmation of iCAFs, how many patients were screened”. All the immunofluorescence staining assays in this study were performed on a tissue chip including 30 paired tumor and normal tissues. iCAFs were found in these samples (**Figure 3C**). In addition, we performed FACS sorting on another 5 tumor samples, also confirmed the existence of iCAFs in bladder cancer tissues (**Figure 4F**).

No matter in scRNA-seq or FACS sorting results, percentage of iCAFs show high variation between samples (scRNA-seq: $7.2\pm 8.2\%$, FACS: $37.5\pm 11.7\%$). Considering that these cells mainly located in the tumor stroma site in bladder cancer tissues (**Figure 3J**), variation between patients may be caused by different tumor purity.

5) With such patient variability even after normalizing for batch effect, number of 8 patients is too small of a sample size to obtain biological significance. It's not surprising to find statistical significance with a data set screen of 52721 cells considering each cell has 46000 protein encoding genes. It's not clear how p-values have been adjusted. Nor is it clear how such a small heterogeneous patient cohort can be interpreted. To be clear, the reviewer does not dispute the validity of the data in the paper, but is concerned that statistical results do not necessarily reflect biological relevance.

Response: Thanks for the reviewer’s suggestion. We agree with the reviewer that a

cohort including 8 patients is too small to investigate the heterogeneity of tumor cells. In this study, different from tumor cells, stromal cells and immune cells integrated well after correction of batch effects (**Figure 1B**). None of the cell subgroups except epithelial cells show patient specificity. It indicated that these cells show homogeneity across patients. And it also indicated that the extremely high heterogeneity of epithelial cells between samples were truly have biological significance. Additionally, via inferCNV pipeline, we attributed the difference to patient specific CNV status (**Figure 1C-1D** and **Supplementary Figure 3A**), which was supported by previous study based on WES that CNVs are highly varied between bladder cancer patients (**Robertson, A.G., et al.**).

We admit that the epidemic significance of a cohort consist of 8 patients is limited. In order to further confirm our finding, we used a tissue chip consisting of 30 paired tumor and non-malignant bladder tissues (**Figure 1G, 3C & 3J**). We will expand the scRNA-seq cohort in the future.

Reference:

Robertson, A.G., et al. Comprehensive Molecular Characterization of Muscle-Invasive Bladder Cancer. *Cell* **171**, 540-556.e525 (2017).

Reviewer #2

Major comments:

Point 1. The bulk of the paper is based on single-cell RNA-seq data and contains only few validation experiments. Nevertheless, the dataset here can be invaluable for the larger research community as it could aid in developing novel hypotheses or be used to cross validate with their own datasets. Therefore I have a number of specific comments or suggestions which are needed to improve the usability of the presented work:

1.1 The authors should refrain from using strong language when their results are

based on correlative or associated conclusions and not validated by an additional experiment or technology. Hence, I would recommend the authors to be more prudent in their conclusions or alternatively provide additional experiments that could validate their novel findings.

Here are just a few examples:

- Lines 10-13: We identified 19 different cell types in the BC microenvironment, suggesting that the downregulated immunogenicity of cancer cells contributes to the formation of an immunosuppressive microenvironment.

It's a big step to link the presence of multiple cell types to a immunosuppressive microenvironment, especially since there is a large variability in cell type composition between different tumor samples.

- Lines 13-14: We also found that monocytes underwent M2 polarization in the tumor region and differentiated.

This statement is based purely on a trajectory analysis and for which – at least – additional analyses should be performed as further explained in point 2.4 below.

- Lines 15-16: Furthermore, the LAMP3+ DC subgroup, which has not been described in BC,

recruited regulatory T cells, leading to the formation of an immunosuppressive TME.

This statement is based entirely on other literature and not proven in the manuscript.

I've limited the examples to the abstract only, but I would encourage the authors to carefully read the other parts of the paper as well.

Response: Thanks for the reviewer's suggestions. We admit that we have used too strong language in our manuscript. The inappropriate expression the reviewer pointed out has been modified now. In addition, we have also checked the whole manuscript and modified similar inappropriate expression.

1.2 Make the biological message of the paper straightforward and more clear by

focusing more on one specific aspect. Ideally, this should be on the effect of the inflammatory cancer-associated fibroblasts on tumor cells, which they suggest is mediated through the inferred signaling pathways. A few simple experiments in line with Figure 4G could already answer several key questions:

Is the iCAF-tumor effect contact dependent or not? Is there a difference?

How important is one individual molecule (ligand or receptor) or is there redundancy between them?

Response: Thanks for the reviewer's suggestion. As shown in **Figure 4F-4G**, we collected iCAFs by FACS sorting and cocultured iCAFs with 2 bladder urothelial carcinoma cell lines, T24 and EJ. This assay was performed in a 6-well transwell apparatus with a 0.4 μm pore size (Corning Incorporated, NY, USA). Bladder cancer cells (2×10^3) were seeded in the lower chamber and iCAFs (1×10^5) were seeded in the upper chamber. Since fibroblast was not possible to pass a transwell apparatus with such a small pore, our result revealed that the iCAF-tumor effect was not contact dependent. In addition, we noticed that iCAFs expressed various growth factors, chemokines and cytokines, and that may account for the pro-tumor effect of iCAFs. In the presenting work, we used CellphoneDB 2 to construct a regulatory network centered by iCAFs, and revealed the potential communication between iCAFs and cancer cells. However, this computation model was not able to specify the effect to one individual molecule. Instead, it provided a list of potential secreting proteins that promote proliferation of tumor cells. We will further screen these molecules and investigate their biological functions in the future.

1.3 Make it easier for others to access and explore the data. Provide scripts to reproduce the results and/or create a graphical user interface to browse through the data. The latter can be done relatively easy with an R Shiny app.

Response: The reviewer's suggestion has been well taken. We agree with the reviewer that a graphical user interface could help other team focusing on cancer biology of bladder carcinoma in the future. Since then, .Rds files of the whole dataset

and major cell types have been created, respectively. These files could be read by a website tool (<https://mbolisetty.shinyapps.io/CellView/>) or in R environment by Shiny R package. These .Rds files will be uploaded along with the raw sequencing data to the public database.

Point 2. The authors use a large repertoire of established computational methods to describe their obtained single-cell RNA-seq data, however details are majorly lacking. Since the obtained scRNA-seq results constitute the majority of the results the authors should do a significant effort to describe each method and decision in more detail. For example, which parameters did they use or test? Did they use data-driven or prior knowledge into their decision making? What was the rationale to use a specific method?

Response: Thanks for the reviewer's suggestion. We agree with the reviewer that each method and decision should be described in more details. The reviewer's suggestion has been well taken. Functions and parameters we used in this pipeline have been detailed in Materials and Methods now.

More specifically I have the following questions:

2.1 Since the authors use the 10X genomics platform to generate their single-cell RNAseq data it would be of interest to know which (or if) samples were processed in parallel? This is important since in their first step they use a method to integrate different datasets. They claim this is necessary to reduce 'batch' effects, however if (some) samples are indeed processed in parallel on this platform, then there should be only a minimal 'batch' effect and the authors might instead remove important biological variability. At minimum a rationale should be provided in the paper as to why they followed this approach.

Response: Thanks for the reviewer's suggestion. Limited to the arrangement of surgery, we could only obtain one tumor or paired tumor and normal mucosa every

time. Since the 10X Genomics platform has high demands on cell viability, and in order to keep the samples fresh, samples from 8 different patients were loaded in batches. Hence, patient number has been used as batch and potential batch effect between different patients was regressed out with IntegrateData function offered by Seurat v3. We are sorry for omitting this information and have detailed in the **Materials and Methods** now.

2.2 Please provide which underlying commands and parameters in the Seurat pipeline have been used. Provide the default values if they have been used. Why did they decide on using 3000 variable genes and how did they determine the number of principal components to use downstream? How did they determine the resolution of the (graph-based Louvain?) clustering? When they subset a cell population, do they run the same exact pipeline again or is this different each time? Importantly, the command to integrate multiple datasets does not remove ‘batch effects’ per se. It identifies mutual nearest neighbors in lower dimensions to adjust the gene expression matrix, in other words it cannot discriminate between biological or technical variation in that process. As such I would suggest the authors to consider not to use the term ‘batch effect’ in this situation.

Response: The reviewers suggest providing underlying commands and parameters in the Seurat pipeline that have been used. The reviewer’s suggestion has been well taken. We have detailed the information in **Methods** now.

The reviewer asked that why we decided on using 3000 variable genes and how we determined the number of principal components to use downstream. Team of Satija R has described the standard workflow for Seurat v3 in their previous work (**Stuart, T., et al.**). They recommend to use top 2000 variable features and 30 top principle components when running IntegrateData function. However, they also suggest maintaining more variable features depending on request. In fact, as shown in **Supplementary Figure 1A**, no matter 2000 or 3000 variable features are both able to correct deviation caused by batches in this cohort. Considering that too few variable features may eliminate biological difference at cell subgroup level, we chose 3000

variable features in this pipeline. We also performed ElbowPlot function of Seurat (**Supplementary Figure 1B**), which also support using top 30 PCs for downstream analysis. Cell clusters were identified with FindCluster function of Seurat V3, the default resolution of which has been set at 0.8. This parameter was set the same at every step. Subsequently, we performed FindAllMarkers function of Seurat to find top markers of every clusters to map them to known cell types (**Supplementary Figure 2A**). We subset the whole data based on this annotation. We have detailed the information of this pipeline in the **Methods** section now. In addition, the markers we used in this pipeline, and top markers we find in this presenting work has been listed in Supplementary Materials now. We hope this will help other teams investigating the TME of bladder carcinoma.

The reviewer expressed their concern that the command to integrate multiple datasets cannot discriminate between biological or technical variation. We agree with the reviewer's opinion. In this study, instead of fastMNN function, we used Seurat V3 to integrate potential batch effect between different samples. Different from fastMNN strategy of scater package, Seurat V3 recommended a new algorithm to integrate different datasets (**Stuart, T., et al.**). The authors call it "anchor". This algorithm was performed on a matrix of variable features and construct a corrected matrix for graph-based learning (tSNE or UMAP), which means that expression level of all the genes was not changed. This strategy could maintain the biological difference maximally.

Reference:

Stuart, T., *et al.* Comprehensive Integration of Single-Cell Data. *Cell* **177**, 1888-1902.e1821 (2019).

2.3 Which cells were exactly used as control group to infer copy number variations? These should also be epithelial (-like) cell types, as this type of analysis is based on gene expression changes and thus the control cell type could have a major influence on the obtained results.

Response: Thanks for the reviewer's suggestion. We agree with the reviewer that control group of InferCNV pipeline should be epithelial cell types. After clustering of CD326+ (EPCAM+) epithelial or epithelial like cells, we chose two single cells clustering almost only detected in non-malignant tissues as the control group. We have detailed this information in the **Methods** now.

2.4 What prior information was used to focus only monocytes and TAMs in the pseudotime trajectory analysis? The authors should apply RNA velocity (La Manno et al) on all the populations within the myeloid populations (Figure 2A) to identify their inter-relationships (and direction).

Response: The reviewer asked what prior information was used to focus only monocytes and TAMs in the pseudotime trajectory analysis. Previously, **Kim, I.S., et al.** has described the monocyte-M2 macrophage polarization and validated in a murine breast cancer model. CD1C+ DCs are also known as myeloid-derived DCs. This subgroup is derived the myeloid lineage, too. However, it was another differentiation direction. **Zhang, Q., et al.** described similar phenomenon in human hepatocellular carcinoma by single cell RNA sequencing. Although CD1C+ DCs show closer distance to monocyte and macrophage, it was clustered into another lineage (**Zhang, Q., et al.**). Since then, we only used monocytes and TAMs in the pseudotime trajectory analysis.

We agree with the reviewer that RNA velocity may better describe the relationship inside myeloid population. As shown in **Supplementary Figure 6B**, in the t-SNE plot of RNA velocity analysis, monocytes show potential to differentiate into TAMs. Similar phenomenon was observed by other teams (**Zhang, Q., et al.,** Guilliams, M., *et al.*) in their previous work.

Reference:

Kim, I.S., *et al.* Immuno-subtyping of breast cancer reveals distinct myeloid cell profiles and immunotherapy resistance mechanisms. *Nature cell biology* **21**, 1113-1126 (2019).

Zhang, Q., *et al.* Landscape and Dynamics of Single Immune Cells in Hepatocellular Carcinoma. *Cell* **179**, 829-845.e820 (2019).

Guilliams, M., Mildner, A. & Yona, S. Developmental and Functional Heterogeneity of Monocytes. *Immunity* **49**, 595-613 (2018).

2.5 For the SCENIC analyses it would be useful to provide some supplemental figures illustrating the enriched regulons (TFs □ downstream genes).

Response: Thanks for the reviewer's suggestion. The reviewer's suggestion has been well taken. The expression level of the downstream genes of top regulons of the monocyte-macrophage lineage have been shown in **Supplementary Figure 6A**. In addition, a list of potential enriched regulons have been provided in **Supplementary Table 2**. We hope the information are useful for other teams.

2.6 Please explain how batch effects were removed with the sva package? Do all the arrays have common samples, which were then used to remove potential technical batch effects or how was this analysis performed? Is this really batch effect removal or “estimating surrogate variables for unknown sources of variation” (sva package)? Please, make sure to use the proper terminology.

Response: The reviewer asked how batch effects were removed with the sva package. We collected all the microarray based datasets available in GEO and ArrayExpress database. Cohort with less than 15 tumor samples were not included in this study. In order to make these samples comparable, all the arrays only contain invasive or non-invasive bladder urothelial carcinoma and non-malignant bladder mucosa, while sample of other histological subtype were not involved in this investigation. We have further detailed this pipeline in the **Methods**.

The reviewer asked that ‘Is this really batch effect removal or “estimating surrogate variables for unknown sources of variation” (sva package)’. As shown in **Supplementary Figure 8B**, different cohorts show significant batch effect. Since then,

we thought that SVA function, which was designed for estimating surrogate variables for unknown sources of variation, was not suitable for this situation. Instead, another algorithm, Combat, of sva package was used to regress potential batch effect out. It is the most prevalent strategy for correcting potential batch effect between microarray-based datasets (Kamoun, A., *et al.*).

Reference:

Kamoun, A., *et al.* A Consensus Molecular Classification of Muscle-invasive Bladder Cancer. *European urology* **77**, 420-433 (2020).

Minor comments:

- **Please provide technical details about the single-cell RNAseq quality (histograms, statistics, ..)**

Response: The reviewer suggested providing technical details about the single-cell RNA seq quality. The reviewer's suggestion has been well taken. Cell numbers captured per sample have been shown in **Supplementary Figure 3B**. Violin plot and tSNE plot showing UMI counts, number of genes and percent of mitochondrial derived transcripts of profiled single cells has been shown in **Supplementary Figure 1C & 1D**. In addition, clinical information of the 8 patients were addressed in **Supplementary Table 1**.

- **Suppl. Fig. 2 and Fig. 1 Are any of the tumor and para-tumor samples matched? If so, did you use that information in your analysis?**

Response: The reviewer asked if any of the tumor and para-tumor samples are matched.

In this cohort, 3 para-tumor samples were obtained from 3 of the 8 patients. We have addressed that information in **Supplementary Table 1**. Since then, we used patient number, instead of sample number to correct potential batch effect.

- Is IGF2 found on the common gained CNVs? Can the authors provide a list of genes that are on the common gained and deleted CNV regions?

Response: The reviewer asked if IGF2 was found on the common gained CNVs. IGF2 was not on the common gained CNVs of bladder cancer. **Knowles, M.A. et. al.** has reviewed the common CNVs of bladder cancer while IGF2 was not included. It means that the up-regulation of IGF2 may be caused by another mechanism, which we will further investigate in our future work.

The reviewer suggested providing a list of genes that are on the common gained and deleted CNV regions. The reviewer's suggestion has been well taken. A list of a list of genes that are on the common gained and deleted CNV regions has been addressed by in their previous review. We must explain that InferCNV pipeline predicts potential CNVs based on single cell transcriptome by gene step-size (<https://github.com/broadinstitute/inferCNV>). It could only predict CNVs in general, which means it may be better to only use it to distinct potential cancer cells from non-malignant cells. In order to identify accurate CNVs at single gene level, single cell DNA sequencing may be a better choice. Since high throughput scDNA-seq technology is still not available now, we still recommend to use the list addressed by **Knowles, M.A. et. al.** at present.

Reference:

Knowles, M.A. & Hurst, C.D. Molecular biology of bladder cancer: new insights into pathogenesis and clinical diversity. *Nature reviews. Cancer* **15**, 25-41 (2015).

- “Among the 3 DC subgroups, LAMP3+ DCs expressed various genes encoding 120 cytokines, including CCL17, CCL19 and CCL22 (Figure 2H)”

The figure call out is most likely wrong and should be Figure 2H.

Response: Thanks for the reviewer's suggestion. That is a writing mistake and we are

sorry to make that. Instead, it should be **Figure 2F** and has been corrected now.

- Figure 2H. Correlation is not an ideal estimator here. The correlation (r) result is likely dependent on the outliers as most of the datapoints are in the lower left part of the graph. Is this pearson or spearman? How are the signatures created?

Response: Thanks for the reviewer's suggestion. R value of correlation result was calculated by Spearman correlation analysis. Differentially expressed genes with FC > 2 were considered as marker genes. Mean TPM value of marker genes was log₂ normalized to create the signature value. We have detailed the method in **Methods** section. Similar strategy has been described by **Zhang, Q., et al.**

Reference:

Zhang, Q., *et al.* Landscape and Dynamics of Single Immune Cells in Hepatocellular Carcinoma. *Cell* **179**, 829-845.e820 (2019).

- There is no Figure 4I and no Figure 4H

Response: Thanks for the reviewer's suggestion. The mistake has been corrected now.

- Can the authors discuss why both iCAF and mCAF display a similar negative association with overall survival in Figure 5A? This does not seem to be in line with their other data.

Response: The reviewer asked that why both iCAF and mCAF display a similar negative association with overall survival in **Figure 5A**. In fact, iCAF and mCAF show more similarity than difference. Since then, it might be difficult for Cibersortx to accurately recognize the percent of these two different fibroblasts in bulk RNA sequencing data. That was restricted by the algorithm. However, as shown in **Supplementary Figure 7F**, iCAF specific gene, PDGFRA is associated with worse OS in TCGA BLCA cohort, while mCAF specific gene, RGS5 is not. Since then, we

thought that iCAF could have more important role in bladder cancer and focused on this subgroup of fibroblast. We must admit that our work does not exclude the potential role that mCAF have in the progression of bladder cancer. We have discussed that in the **Results** section.

- **“Together, our results predicted that iCAFs could promote the proliferation of tumor cells and stromal cells and could also recruit immune cells into the tumor stage, leading to the establishment and maintenance of an immunosuppressive microenvironment.”**

This statement seems to be contradictory. The authors should explain better how increased recruitment of immune cells and an immunosuppressive microenvironment can co-exist. For example, increase in regulatory T-cells are also observed with strong immune activation, as this is believed to be a normal immunological feedback mechanism.

Response: Thanks for the reviewer’s suggestion. We agree with the reviewer that this inference was rigorous enough. Existing evidence could only lead to the inference that iCAFs take part in the recruitment of immune cells. We have modified the statement now.

- **As a follow-up on the previous comment: in supplemental Figure 2G the tumor samples display a large variability in cell-type composition, which does not seem to be associated with tumor grade. Can the authors comment or discuss what this might suggest in light of their model?**

Response: The reviewer asked that what the variability in cell-type composition of tumor samples might suggest in light of our model. Via correlating scRNA-seq data to bulk RNA seq data, we found that bladder cancer samples of different molecular subtype show variability in cell-type composition. As shown in **Figure 5C & 5H**, luminal papillary subgroup shows the highest tumor purity, which means sample

belong to this group contains more epithelial and less stroma cells. Although most of the low grade samples are classified as luminal papillary, there's still a considerable part of high grade belonging to this subgroup. We thought that could account for why several high grade tumor samples does not show that much stromal cells.

Reviewer #3:

The authors perform single cell analysis based on 8 tumors and 3 non-tumor samples from patients with bladder cancer. Overall, based on the number of tumor samples included, the claims in the paper are too strong. My overall concern is that if 8 new tumors are analyzed, they may show different patterns.

Response: The reviewer's expressed their concern that if 8 new tumors are analyzed, they may show different patterns and the claims in the paper are too strong. We agree with the reviewer's opinion and have modified these inappropriate expressions in this paper.

We agree with the reviewer that 8 samples are not enough to investigate heterogeneity of bladder carcinoma. Since then, we used a tissue chip including 30 paired bladder tumor and non-malignant tissues to validate our finding at protein level. In addition, we used several bulk sequencing databases to cross-validate our findings.

In order to make the results more convincing, we mainly focused on the non-malignant cell types, such as myeloid cells and fibroblasts. These cells show more homogeneity than heterogeneity between patients.

In the future, we will expand the cohort and further investigate the heterogeneity of cancer cells.

Some specific points:

1. Line 31: Not correct. Significant progress have been made in therapeutic strategies for bladder cancer patients in recent years.

Response: Thanks for the reviewer's suggestion. We have modified this incorrect statement now.

2. Line: 80: I don't think the cancer cells are producing immunoglobulins – this claim seems strange

Response: Thanks for the reviewer's suggestion. As shown in **Figure 2D**, cancer cells do not produce immunoglobulins indeed. Instead, these immunoglobulin coding genes were detected in normal bladder epithelial cells. We meant to express that comparing to non-malignant epithelial cells, cancer cells don't express immunoglobulin coding

genes, and express less MHC-II molecules. We have taken the reviewer's suggestion and modified the statement now.

3. Lline281: patient samples, what kind of tumor stages are analyzed here? Outcome? is it comparable to the MIBC TCGA data?

Response: The reviewer asked what kind of tumor stages are analyzed here. The clinical information has been addressed in **Supplementary Table 1**. 4 invasive and 4 non-invasive samples were included in this analysis. Since these samples were obtained and profiled recently, these patients are all alive until now. The outcome of these patients is still unknown.

The reviewer asked if this scRNA-seq cohort is comparable to the MIBC TCGA data. In this bioinformatic pipeline, we noticed that although the tumor cells show high heterogeneity between patients, other stromal cells, including immune cells, fibroblasts and endothelial cells show relative homogeneity (**Supplementary Figure 2C**). Difference between patients was not caused by different cell subgroups, but different percent of similar cell subgroups. Since then, we chose not to divide epithelial cells into subgroups in the Cibersortx pipeline. We thought this algorithm could reveal the constitution of TME of MIBC TCGA cohort.

4. In Figure 1C it looks like most of the iCAF cluster is driven primarily by one patient. In figure 3 this looks different – please explain.

Response: Thanks for the reviewer's suggestion. iCAFs were identified in almost all samples in this cohort. This mistake was caused by colors of the **Figure 1B**. We are sorry for that and have changed the color, which may make it clearer. Additionally, **Figure 3A** was colored by cell cluster numbers generated by machine learning, that information has been addressed in the figure legend now.

5. Figure 1D – difficult to interpret this, what is actually shown in the different expression clusters (pathway enrichment etc). Overall, many of the figures (and the logistics behind) are difficult to follow. E.g. different cluster numbers are

used (beginning from 0 and beginning from 1 – how should one follow this?). Also why are e.g. the specific clusters in 2c selected, and why are only two cluster selected in 2E.

Response: Thanks for the reviewer's suggestion. We agree with the reviewer that pathway analysis should be performed. We noticed that genes in the red blank of **Figure 1D** was significantly down-regulated in bladder cancer samples. Since then, we performed GO enrichment analysis on these genes (**Figure 1E**), finding out that these genes were enriched in immune-related pathways.

The reviewer expressed that many of the figures (and the logistics behind) are difficult to follow. We used the same number to number cell clusters of one lineage. For example, as shown in **Supplementary Figure 5A**, myeloid lineage was divided into 13 different subgroups, which was numbered from 0-12. Subsequently, we noticed that cluster 0,1,3,5 and 9, which was identified as monocyte and macrophage, show potential differential relationships (**Figure 2B, red blank**). Similar phenomenon was recently observed by another team in hepatocellular carcinoma (**Zhang, Q., et al.**). Since then, only these 5 cell clusters were used in the trajectory analysis. That why we did not number them from cluster 1. In order to make the statement clearer, we have added this information to the figure legend of **Figure 2**. Thanks for the reviewer's suggestion.

The reviewer asked that why are e.g. the specific clusters in 2c selected, and why are only two cluster selected in 2E. **Figure 2E** was the results of trajectory analysis, which shows the continual change of expression level of these immune checkpoints. The cluster shown in **Figure 2E** was the gene clusters returned by **plot_pseudo_heatmap** function of Monocle 2 package. The cluster means that these genes show 2 different changes in the differentiation process. We have added the information to **Methods** and figure legends of **Figure 2E** now.

Reference:

Zhang, Q., *et al.* Landscape and Dynamics of Single Immune Cells in Hepatocellular Carcinoma. *Cell* **179**, 829-845.e820 (2019).

REVIEWERS' COMMENTS:

Reviewer #1 (Remarks to the Author):

The revised version of manuscript looks good to me. No more questions to authors.

Reviewer #2 (Remarks to the Author):

The authors are to be commended for their efforts to improve the writing of the paper, adding detailed information for the conducted analysis and providing an interactive tool to make explorative analysis easier for the research community. Nevertheless I do have two minor comments which I believe would improve the overall quality of the manuscript.

1) The monocyte to TAM conversion seems to be mainly supported by previous research. The RNA velocity analysis seems to suggest that there might be some differentiation in that direction, but that monocytes seem to mostly mature/differentiate in another direction (see monocyte cluster in Suppl Fig. 6B) . Therefore I believe that they should make it more clear that this analysis is primarily based on previous data and known biology.

2) The current discussion is mostly a repetition of the conclusions in the main text. I think the manuscript would benefit from providing a more in depth discussion about some of the results or current limitations with this type of research. For example:

- a. Current scRNAseq experimental designs and tools to remove 'batch' effects will still remove patient specific information when patient information and sequencing batch is completely confounded. This is currently common practice since long-term storage of samples often results in decreased data quality, however it's important to understand this current limitation.
- b. What are the limitations of the ligand-receptor analysis and what could be done in follow up experiments. Would spatial information be more informative?
- c. What are the limitations of deconvolution approaches? This is relevant for their main message as they show that it is hard to discriminate between the mCAF and iCAF signature.
- d. ...

Editorial note for the authors:

Confidentially, Reviewer #2 has provided additional comments on the R Shiny app. He has highlighted that the Shiny App could be run successfully in a local browser, but that errors happen when web browser are used (both with Firefox and Chrome). He has suggested to fix this issue and also to provide additional information for novice R users on how to locally run the Shiny app. Alternatively, he suggests to create a small wrapper script that installs all the necessary R packages and launches the shiny app.

Finally, he also suggests the authors to make sure that the figures presented in the manuscript correspond with those presented in the Shiny app (i.e. dimension reduction coordinates and available annotation), to make the use of the app together with the paper easier. As an example, he mentions that the myeloid clusters in the manuscript have all been given distinct names, but that they are represented as integers on the Shiny app".

Reviewer #3 (Remarks to the Author):

[no further comments]

To reviewer 2:

The authors are to be commended for their efforts to improve the writing of the paper, adding detailed information for the conducted analysis and providing an interactive tool to make explorative analysis easier for the research community. Nevertheless I do have two minor comments which I believe would improve the overall quality of the manuscript.

1) The monocyte to TAM conversion seems to be mainly supported by previous research. The RNA velocity analysis seems to suggest that there might be some differentiation in that direction, but that monocytes seem to mostly mature/differentiate in another direction (see monocyte cluster in Suppl Fig. 6B). Therefore I believe that they should make it more clear that this analysis is primarily based on previous data and known biology.

Response: Thanks for the reviewer's suggestion. We highly agree with the reviewer's opinion. M2 polarization is a confirmed phenomenon across cancer types. Previously, mainstream view was that these cells were polarized from M1 cell into M2 cells, while recently it was reported that monocytes were recruited directed into the tumor region and then polarized into M2 stage in absent of the M1 stage (Kim, I.S., *et al.*). Single cell level works have also been published by **Zhang, Q., et al.** We have modified the statement now.

Reference:

Kim, I.S., *et al.* Immuno-subtyping of breast cancer reveals distinct myeloid cell profiles and immunotherapy resistance mechanisms. *Nature cell biology* **21**, 1113-1126 (2019).

Zhang, Q., *et al.* Landscape and Dynamics of Single Immune Cells in Hepatocellular Carcinoma. *Cell* **179**, 829-845.e820 (2019).

2) The current discussion is mostly a repetition of the conclusions in the main text. I think the manuscript would benefit from providing a more in depth discussion about some of the results or current limitations with this type of research. For example:

- a. Current scRNAseq experimental designs and tools to remove ‘batch’ effects will still remove patient specific information when patient information and sequencing batch is completely confounded. This is currently common practice since long-term storage of samples often results in decreased data quality, however it’s important to understand this current limitation.
- b. What are the limitations of the ligand-receptor analysis and what could be done in follow up experiments. Would spatial information be more informative?
- c. What are the limitations of deconvolution approaches? This is relevant for their main message as they show that it is hard to discriminate between the mCAF and iCAF signature.
- d. ...

Response: The reviewer’s suggestion has been well taken. We have modified the discussion part to make it more in-depth.